# Tracking the coherent generation of polaron pairs in conjugated polymers

Antonietta De Sio[1,2], Filippo Troiani[3], Margherita Maiuri[4], Julien Réhault[4], Ephraim Sommer[1,2], James Lim[5], Susana F. Huelga[5], Martin B. Plenio[5], Carlo Andrea Rozzi[3], Giulio Cerullo[4], Elisa Molinari[3,6] & Christoph Lienau[1,2,7]

The optical excitation of organic semiconductors not only generates charge-neutral electron-hole pairs (excitons), but also charge-separated polaron pairs with high yield. The microscopic mechanisms underlying this charge separation have been debated for many years. Here we use ultrafast two-dimensional electronic spectroscopy to study the dynamics of polaron pair formation in a prototypical polymer thin film on a sub-20-fs time scale. We observe multi-period peak oscillations persisting for up to about 1 ps as distinct signatures of vibronic quantum coherence at room temperature. The measured two-dimensional spectra show pronounced peak splittings revealing that the elementary optical excitations of this polymer are hybridized exciton-polaron-pairs, strongly coupled to a dominant underdamped vibrational mode. Coherent vibronic coupling induces ultrafast polaron pair formation, accelerates the charge separation dynamics and makes it insensitive to disorder. These findings open up new perspectives for tailoring light-to-current conversion in organic materials.

[1] Institut für Physik, Carl von Ossietzky Universität, Oldenburg 26129, Germany. [2] Center of Interface Science, Carl von Ossietzky Universität, Oldenburg 26129, Germany. [3] Istituto Nanoscienze—CNR, Centro S3, via Campi 213a, Modena 41125, Italy. [4] IFN-CNR, Dipartimento di Fisica, Politecnico di Milano, Milano 20133, Italy. [5] Institut für Theoretische Physik and IQST, Universität Ulm, Ulm 89069, Germany. [6] Dipartimento di Scienze Fisiche, Matematiche e Informatiche, Università di Modena e Reggio Emilia, via Campi 213a, Modena 41125, Italy. [7] Research Center Neurosensory Science, Carl von Ossietzky Universität, Oldenburg 26111, Germany. Correspondence and requests for materials should be addressed to C.L. (email: christoph.lienau@uni-oldenburg.de).

Thin films of conductive polymers combine mechanical flexibility with excellent optical and electronic properties and hence are key active materials for flexible organic optoelectronic devices such as solar cells[1,2], field effect transistors[3–5] or light emitting diodes[6–8]. Such organic π-conjugated semiconductors have striking and rather complex optical properties. Their optical excitation not only results in the formation of excitons, Coulomb bound electron-hole pairs, but—in contrast to, for example, inorganic semiconductors—also in the creation of a wealth of other quasiparticles, namely polaron pairs, polarons, biexcitons and others[9–11]. Polaron pairs, sometimes also called spatially indirect or charge-transfer excitons, are charge-neutral excitations in which spatially separated electrons and holes weakly interact via their Coulomb attraction and are each coupled to their own lattice distortion. They are thought to act as precursors of free charges, which in polymers are also coupled to the lattice and appear in the form of electron and hole polarons[11,12]. Hence, both polaron pairs and polarons are of fundamental importance, for instance, for the efficient light-to-current conversion in polymer-based photovoltaic devices.

High-yield formation of polaron pairs in different types of neat polymers, including poly-phenylene-vinylene[13], poly(3-hexylthiophene) (P3HT)[14–18] and low bandgap copolymers[19] has been reported by several groups. Interestingly, all existing time-resolved spectroscopic studies indicate that both excitons and polaron pairs are simultaneously created within the time resolution of the experiment, typically few tens of femto-seconds[15,20]. This puzzling observation has led to a substantial debate about the physical mechanisms underlying ultrafast polaron pair formation. Different scenarios are commonly invoked, including the independent photogeneration of excitons and polaron pairs[20], the rapid formation of polaron pairs by exciton dissociation[21] or, polaron pair formation from delocalized coherent excitations[22]. Recent theoretical work[23,24] suggests that strong coherent coupling between electronic and vibrational degrees of freedom, that is, vibronic coupling, may lie at the origin of this efficient polaron pair formation, similar to seminal models for polaron motion in solids[25]. Such a picture would underpin emerging experimental and theoretical evidence that vibronic couplings are important for photoinduced energy and charge transfer processes in biological and artificial light-harvesting systems[26–30] or organic solar cells at room temperature[31–33]. Yet, at present, it is unclear to what extent vibronic quantum coherence[34–37] affects the optical properties of thin polymer films at room temperature. Very recently, two-dimensional electronic spectroscopy (2DES) has been used to study vibrational coherence in P3HT polymer thin films[18]. In experiments performed on non-annealed samples, polaron pair absorption has been observed but no detailed insight into their formation dynamics could be obtained. Here, we use 2DES with 10-fs temporal resolution to dynamically probe polaron pair formation in annealed regioregular (rr-) P3HT thin films at room temperature. We observe multi-frequency time-domain oscillations and spectral peak splittings as clear signatures of strong vibronic coupling between excitons, polaron pairs and a dominant underdamped vibrational mode of the polymer.

## Results

**Linear absorption spectrum.** We prepare our samples by spin coating from chlorobenzene and subsequent annealing (see Methods) according to a recipe used for the fabrication of efficient organic solar cells based on this polymer[38]. In this way, we ensure that the film morphology is similar to that used in device applications. The linear optical absorption spectrum of P3HT (Fig. 1a, red line) shows weak, predominantly inhomogeneously broadened vibrational progression with peaks at 2.07, 2.25 and 2.41 eV. Commonly, the absorption spectrum is described with a model[39] in which this vibrational progression reflects transitions from the lowest vibrational state in the electronic ground state $|G, 0\rangle$ to different vibrational states $|v\rangle$ in the excited state exciton manifold $|X, v\rangle$. Both ground and excited states are assumed to be coupled to a single vibrational mode, the symmetric $C=C$ stretch mode of the thiophene ring at $1,450\ cm^{-1}$ ($\sim 180$ meV) with a vibrational period of 23 fs. In this incoherent exciton model, nearest neighbour excitonic coupling with strength $J$ and site disorder are included, resulting in a delocalization of the excitonic wavefunction over 20–25 monomer units[39]. All other photoexcitations, such as polaron pairs and polarons, are formed due to incoherent population relaxation from the optically excited exciton state. In the limit of vanishing $J$, the model accounts well for the low energy part of the linear absorption (Fig. 1a, black line) if a large Huang-Rhys factor of about 1.8 (Supplementary Note 5) is considered for the displacement between ground and exciton potential energy surfaces. Smaller Huang-Rhys factors of $\sim 1$ are deduced for a finite nearest neighbour excitonic coupling[39]. Due to the weak transition dipole moment between ground and polaron pair states, the absorption spectrum does not provide much information on charge-separated states.

**Transient absorption spectra.** The incoherent exciton model can also describe the ultrafast transient absorption spectra of P3HT[18] reasonably well. These spectra reveal a photobleaching of the lowest two vibronic resonances (Fig. 1b) and a photoinduced absorption band around 1.89 eV, that is, at energies below the polymer bandgap[14,15,18,23], which appears on a sub-picosecond timescale (see Supplementary Notes 1–2). This photoinduced band in P3HT is generally assigned to excited state absorption from polaron pair transitions[15,33,40]. However, the assignment of sub-bandgap features in the optical spectra of conjugated polymers is challenging[11]. Bands in a similar spectral region have also been assigned to hole polarons[14,18]. For our film the latter assignment seems unlikely since Terahertz photoconductivity measurements show inefficient photogeneration of free polarons on a picosecond time scale[41,42]. Also, as we will discuss in detail below, the results presented in our work strongly support an assignment of this band to Coulomb-correlated, weakly bound polaron pairs. Therefore, in the following we take this band as a marker for the formation of polaron pairs upon impulsive optical excitation of excitons.

**Ultrafast two-dimensional electronic spectroscopy.** To study the polaron pair formation in more detail, we recorded 2DES maps of the polymer thin film (Fig. 1c). We used a partially collinear pump-probe geometry, pumping the sample with a pair of phase-locked broadband sub-8-fs pulses with variable time delay $\tau$ and probing with a replica delayed by $T$. By recording the differential trans-mission spectrum $\Delta T(\tau, T, E_D)$ and taking the real part of its Fourier transform along $\tau$, we obtain the absorptive 2D map $A_{2D}(E_X, T, E_D)$ as a function of the excitation $E_X$ and detection $E_D$ energies for each waiting time $T$. More details are provided in the Methods.

2DES maps at selected waiting times are shown in Fig. 1c–f. We start by discussing the data recorded for a waiting time of 2.3 fs (Fig. 1c). Surprisingly, in the excitonic region of the spectrum, at detection energies between 1.95 and 2.3 eV, we observe a clear splitting of the 2D map into a multitude of absorptive peaks. In total, we find 16-positive peaks, marked by blue open circles in Fig. 1c. Cross sections along detection and excitation energy, both taken at the first vibronic resonance (2.25 eV), clearly reveal these splittings. They are apparently arranged in four quartets, each centered around the positions of

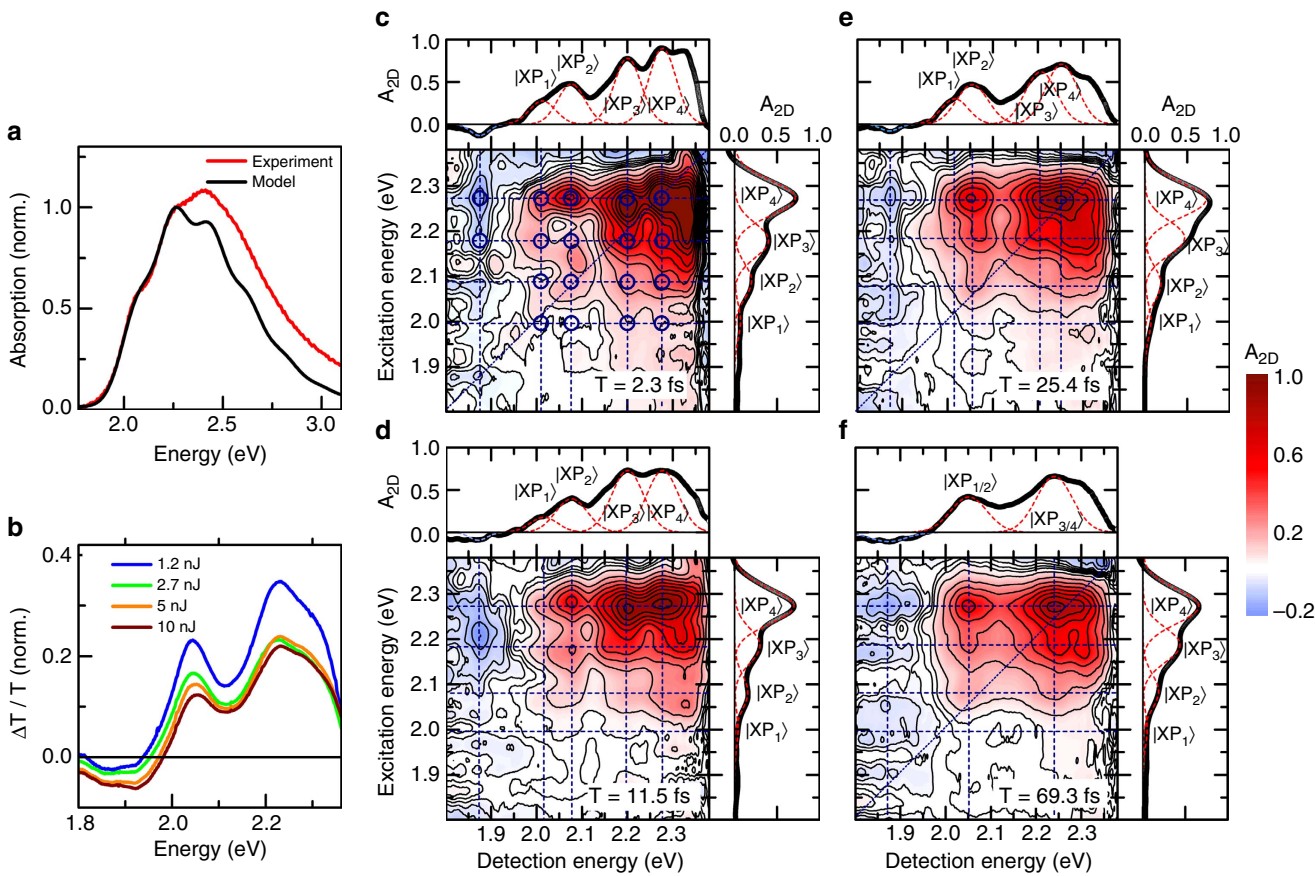

**Figure 1 | Two-dimensional electronic spectroscopy of annealed rr-P3HT thin films.** (**a**) Linear absorption spectrum (red) and simulation (black), based on a displaced oscillator model[39], coupling ground and exciton states to the C=C stretch mode at 1,450 cm$^{-1}$ (23-fs period). (**b**) Differential transmission ($\Delta T/T$) spectra at a delay of 350 fs, normalized to the pump energy. Positive ($\Delta T/T > 0$) bands are due to the bleaching of excitonic transitions, whereas the negative signal around 1.89 eV is assigned to photoinduced absorption of polaron pairs. (**c–f**) Absorptive 2DES maps of annealed rr-P3HT thin films at selected waiting times of (**c**) 2.3 fs, (**d**) 11.5 fs, (**e**) 25.4 fs and (**f**) 69.3 fs. At early waiting times (**c–e**) Cross sections along both the excitation and detection energy reveal distinct splitting of the bleaching peaks into four vibronic resonances, labelled $|XP_1\rangle$ to $|XP_4\rangle$. The corresponding peaks are marked with dark blue circles. These splittings are the characteristic signature of strong vibronic coupling resulting in the formation of hybrid exciton-polaron-pair (XP) modes. The cross peaks with negative amplitude at detection energy of 1.89 eV originate from photoinduced absorption of polaron pairs. At longer waiting times (**f**) the splitting along the detection energy washes out and the resulting cross sections match those of the low-energy vibronic peaks in $\Delta T/T$ spectra. At all waiting times, negative amplitudes are observed for detection energy around 1.89 eV, monitoring polaron pair peak dynamics.

the vibronic resonances $|X, v\rangle$ predicted by the incoherent exciton model. In each quartet, we observe peak splittings of about 80 meV along both the excitation and detection axes. These spectral splittings can also be seen in the three 2D maps at early waiting times shown in Fig. 1c–e and in additional maps shown in Supplementary Fig. 6 at waiting times up to about 60 fs. In these data, we observe that the splitting along the detection axis gradually decreases with increasing waiting time (see Supplementary Note 4). At waiting times longer than 60 fs (Fig. 1f), the substructure along the detection axis washes out and the cross section resembles more that of the inhomogeneously broadened transient absorption spectra (Fig. 1b).

In addition, we observe pronounced negative cross peaks ($A_{2D} < 0$) at detection energy of about 1.89 eV, corresponding to photoinduced absorption of polaron pairs. These negative peaks appear at excitation energies in the exciton region and are already well resolved at early waiting times, directly after photoexcitation (Fig. 1c). According to the incoherent exciton model one would expect peaks centered at excitation energies around 2.25 eV and 2.07 eV, corresponding to the two lowest-energy exciton transitions in the absorption spectrum, respectively. Instead, we observe, around 2.25 eV, a splitting into two excitation peaks

separated again by about 80 meV (Fig. 1c). The signal around 2.07 eV is too weak to be resolved.

The dynamics of selected peaks during the first 170 fs after photoexcitation are shown in Fig. 2b–f (open circles). Dynamics of these peaks for longer waiting times up to 1 ps are reported in the Supplementary Information (see Supplementary Note 3). For all peaks we see pronounced temporal oscillations along the waiting time. These oscillations are well described by a single, exponentially damped cosine function with a period of 23 fs, corresponding to the C=C stretch mode, and a damping time of 650 fs (Fig. 2b–f solid lines and Supplementary Fig. 4). The Fourier transform of these dynamics (Fig. 2b,c, insets) is clearly dominated by the C=C stretch mode[43]. Additionally, we also find weak components at around 500, 1,000 and 2,000 cm$^{-1}$, which do not match any of the known intramolecular vibrational modes of the polymer[43]. Instead, we observe that the peak splitting of about 80 meV that is seen in the 2DES maps at early times, corresponds to a beat frequency of about 600 cm$^{-1}$, approximately matching the low-frequency component in the Fourier transforms (Fig. 1c–f).

Similar oscillatory features are also observed on the negative cross peaks at $E_D = 1.89$ eV, where the negative signal appears rapidly within the first vibrational period (Fig. 2f open circles).

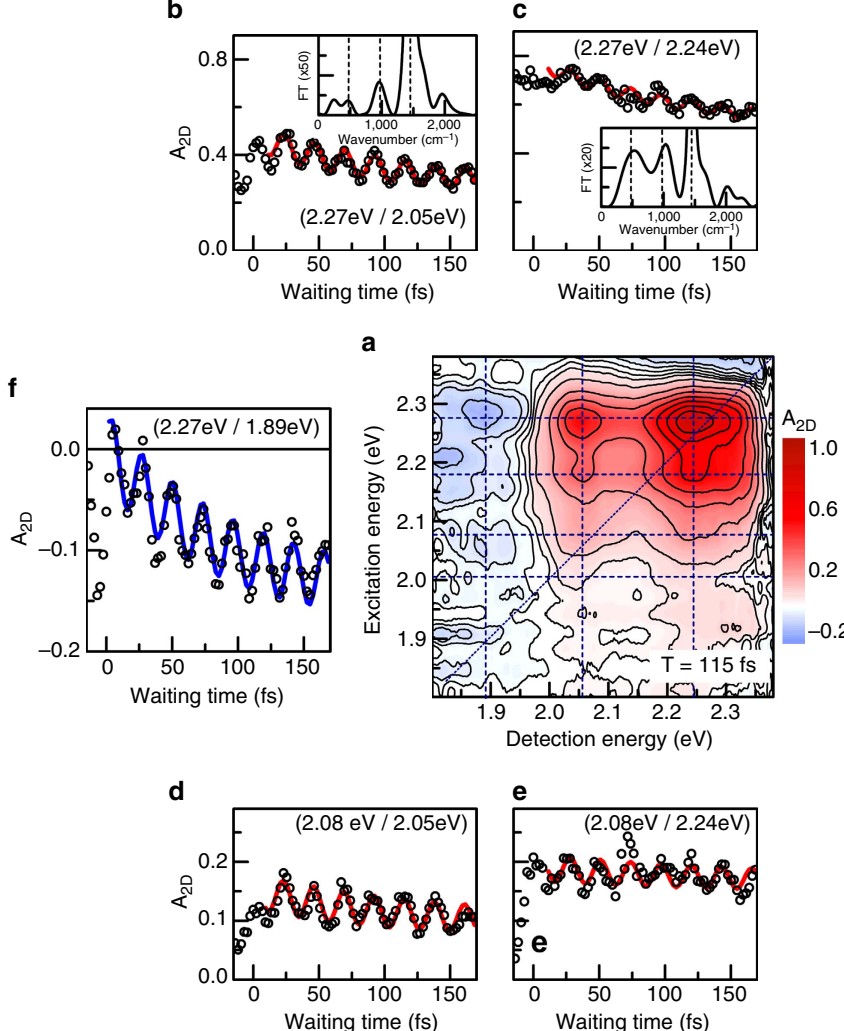

**Figure 2 | Dynamics of relevant peaks in the absorptive 2DES maps.** (**a**) 2DES map at waiting time of 115 fs. We detect a series of positive ($A_{2D} > 0$) bleaching peaks and negative peaks for detection around 1.89 eV, reflecting photoinduced absorption of polaron pairs. The corresponding resonances are marked by dashed lines. (**b–e**) The dynamics of different positive vibronic peaks (open circles) show pronounced, long-lived oscillations, predominantly with 23-fs period (fit: red line). Fourier spectra (insets in **b,c**) reveal additional oscillatory components at 500$^{-1}$, 1,000$^{-1}$ and 2,000 cm$^{-1}$. (**f**) Remarkably, the negative polaron pair peak amplitude (open circles) displays strong oscillations with 23-fs period and rises within 100 fs (blue line). All these features are distinct signatures of coherent polaron pair formation.

Subsequently the 2D signal rises exponentially with a time constant of about 100 fs, superimposed on the long-lived 23-fs-oscillations (Fig. 2f blue line and Supplementary Fig. 4). We take the rapid appearance of the polaron pair signal as a sign of an ultrafast and potentially coherent formation of polaron pairs within < 20 fs.

The unexpected subpeak structure and the additional low-frequency oscillatory features cannot be explained on the basis of the incoherent exciton model[39], which predicts positive diagonal and cross peaks only at the two lowest-energy vibronic resonances, that is, only four-positive peaks. Moreover, the dynamics of these peaks along the waiting time would show a single long-lived oscillatory component at the frequency of the vibrational mode, that is, 1,450 cm$^{-1}$. Our experimental 2DES data instead provide evidence for coherent coupling between exciton, polaron pair and the dominant vibrational mode of the polymer.

## Discussion

To validate this conclusion, we compare our experimental results to quantum-mechanical simulations of 2DES maps based on the

conceptually simplest model capable of describing the most salient experimental findings. To this aim, we consider a dimeric version of the Holstein Hamiltonian[23] including a phonon-induced modulation of electronic coupling[44] between exciton and polaron pair states[24,45–47]. A similar model allowing for a coherent superposition of localized Frenkel and charge transfer excitons has been proposed to model the static absorption spectrum of oligoacene crystals[47], whereas, commonly, this superposition is not considered in models of the static absorption of P3HT[39]. In our model, the system parameters are chosen to approximate experimental signals in linear and two-dimensional spectroscopy. We model the ground and exciton states as two displaced harmonic oscillators where the difference in equilibrium position is quantified by a Huang-Rhys factor of 2, estimated from the linear absorption spectrum when neglecting intermolecular coupling (Fig. 3f). The frequency of the vibrational mode is taken as that of the C=C stretch mode (1,450 cm$^{-1}$). We assume that the neutral exciton state is electronically coupled to a second, neutral polaron pair state with an electronic coupling strength $J$ of about 90 meV. Also the polaron pair state is coupled to the C=C stretch vibration and described by a third displaced

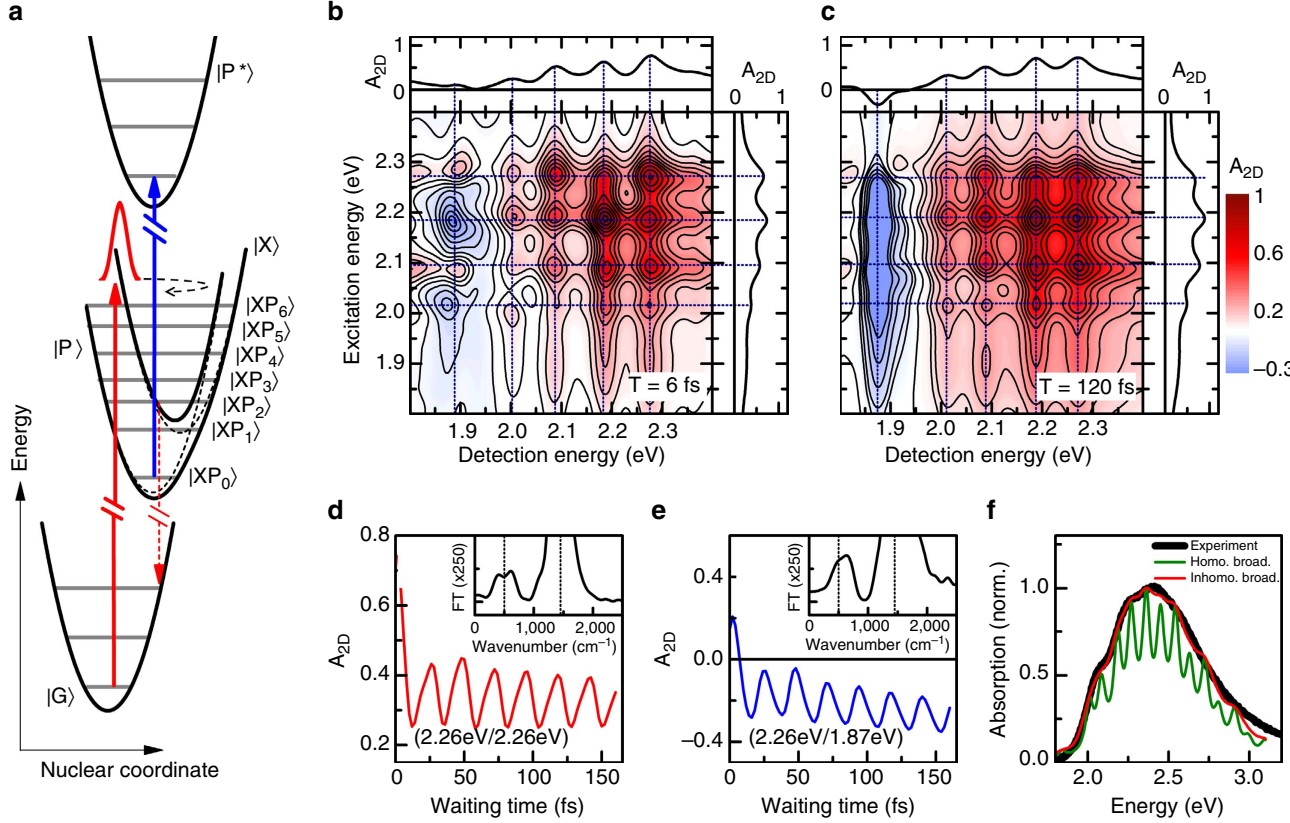

**Figure 3 | Theoretical analysis.** (**a**) Displaced harmonic oscillator model consisting of four electronic levels strongly coupled to the C = C stretch mode. Coherent coupling among exciton $|X\rangle$ and polaron pair $|P\rangle$ and the dominant vibrational mode results in hybridized $|XP_i\rangle$ states. (**b,c**) Simulated absorptive 2D maps in the absence of inhomogeneous broadening at waiting times of 6 and 120 fs, respectively. Positive peaks reflect bleaching of the $|G, 0\rangle \rightarrow |XP_i\rangle$ transitions. The negative peak around 1.89 eV is due to excited state absorption from the $|XP_i\rangle$ manifold to the $|P^*\rangle$ state. Cross sections reveal distinct spectral splitting of the bleaching peaks, in good agreement with the experimental observations (Fig. 2a,b). (**d**) Dynamics of the positive diagonal peak amplitude at 2.06 eV. (**e**) Dynamics of the polaron pair peak for excitation at 2.26 eV and detection at 1.87 eV. A significant part of the polaron pair population is already formed during the first vibrational period. Fourier transforms (**d,e** insets) reveal the vibrational mode at 1,450 cm$^{-1}$ and a weaker component around 500 cm$^{-1}$ arising from electronic coupling between $|X\rangle$ and $|P\rangle$. (**f**) Simulated linear absorption spectrum in the absence (green) and presence (red) of inhomogeneous broadening. For comparison, the black curve depicts the actual experimental data.

harmonic oscillator with a slightly smaller Huang-Rhys factor of about 1.4. We assume that the polaron pair state is energetically detuned by ∼230 meV below the exciton state as sketched in the potential energy surface shown in Fig. 3a. This polaron pair state is optically coupled to an excited polaron pair state. Both states are assumed to couple to the C = C stretch vibrational mode with the same Huang-Rhys factor. In the model simulations, dephasing and relaxation processes are described by the Lindblad formalism[28]. By comparison to the experimental data, we take an electronic dephasing time of about 25 fs for all states and a vibrational relaxation time of 650 fs for the C = C stretch mode. In this model, the photoexcitation of the system by an ultrashort phase-locked pulse pair launches an electronic wavepacket near the inner turning point of the exciton potential energy surface. This wavepacket periodically oscillates back and forth between the coupled exciton and polaron pair states. Its dynamics is monitored by the time-delayed probe pulse, inducing the optical transition between the polaron pair and the excited polaron pair states. For comparison with the experiment, a finite inhomogeneous broadening of the electronic transitions as well as realistic pulse-shapes have been taken into account in the model. In the absence of inhomogeneous broadening, the simulated 2D maps at early waiting times (Fig. 3b) indeed show pronounced splittings of the positive bleaching peaks into a total of 16 sub-peaks. The peak positions match the energy levels of the

hybridized exciton-polaron-pair states. As in the experiment, a negative signal appears at the polaron pair peak (1.89 eV). With increasing waiting time, the sub-peak structure remains (Fig. 3c) and the amplitude of the polaron pair peak increases. Also, the waiting time dynamics of the different sub-peaks predicted by this simple model qualitatively matches that observed experimentally. Specifically we observe pronounced oscillatory modulation of all the bleaching peaks (Fig. 3d) and the polaron pair peaks (Fig. 3e). The latter reveal coherent formation of polaron pairs within half a vibrational period followed by a slower increase in peak amplitude, superimposed on the strong oscillatory modulations. The Fourier transform of both peaks dynamics (Fig. 3d,e insets) shows a main peak centered at the frequency of the C = C stretch mode and an additional peak around 500 cm$^{-1}$. This low-frequency peak and the splitting of the vibronic resonances in the simulations demonstrate that the coherent vibronic model can reproduce the main features observed in the experiments. Hence they provide unambiguous evidence of coherent polaron pair formation in our samples. We also found that these experimental observations cannot be explained by an incoherent model, where population transfer between exciton and polaron pair states is governed by relaxation (see Supplementary Notes 5–8).

Our model simulations confirm that the experimental peak splittings are a direct signature of strong coupling between exciton and polaron pair states. Since the impulsive optical

excitation of the system initially excites excitonic states, this strong coupling induces rapid and oscillatory population transfer between the coupled states[48]. Hence, our results provide independent support for the assignment of the experimental photoinduced absorption feature at 1.89 eV to polaron pairs. The formation of hole polarons necessarily implies a finite spatial separation between electrons and holes that is so large that their Coulomb attraction is negligible. Hence both particles do not interact and it is difficult to conceive a mechanism that drives their coherent and ultrafast geminate recombination. We, therefore, think that the signatures of strong coupling that are seen experimentally very likely reflect the coherent formation of Coulomb-correlated polaron pairs.

When static inhomogeneous broadening is included in the simulations, the sub-peak contrast gradually decreases with increasing inhomogeneity (Supplementary Fig. 8). Remarkably, when choosing a Gaussian broadening of ∼90 meV, similar to the strength of the electronic coupling, the linear absorption spectrum predicted by our phenomenological coupling model is in very good agreement with the experimentally measured spectrum (Fig. 3f). In contrast to the incoherent exciton model[39] (Fig. 1a, black), not only the low-energy region, but basically the lineshape of the complete spectrum is matched by assuming strongly-coupled exciton and polaron pair excitations and their interaction with one dominant underdamped vibrational mode. For this magnitude of the inhomogeneous broadening, the sub-peak splitting of 2DES peaks is still visible (Supplementary Fig. 8), but slightly less pronounced than in the experiment.

P3HT is expected to have a high degree of semi-crystallinity, that is a microstructure exhibiting at the same time highly ordered domains and completely amorphous regions. This picture is confirmed by several studies of this model polymer that have addressed the critical task of understanding and possibly controlling the morphology-function relation of thin films[38,49–52]. From these studies it is known that not only characteristic parameters like the average molecular weight, but also the processing conditions (for example, solvent and annealing) have an impact on the morphology. In particular, all these parameters may change the amount and size of crystallites and amorphous domains[38]. It is also frequently suggested in the literature that this interplay between crystalline and amorphous regions affects the linear optical spectra of P3HT in the sense that at least two types of excitons, that is, one at lower energies from the crystallites and the other one at higher energies from the amorphous regions, contribute to the spectra. The absorption spectrum of excitons in crystalline regions is formally described by the exciton model introduced by Spano[39], which nicely reproduces the low energy part of the absorption spectrum, but not the high energy part (cf. Fig. 1a). According to the literature[38,53], the discrepancy is due to the absorption of excitons from amorphous regions which should result in a higher energy, unstructured band in the linear absorption that is not accounted for in the model. Based on this model[39] one may expect that the persistent vibronic coupling phenomena observed in our experiments can mainly be assigned to excitons in crystalline domains, whereas the amorphous regions may act as blocking layers suppressing charge and excitation transport between adjacent crystallites. In such a scenario any type of vibronic quantum coherence at early times may have little effect on the transport across crystalline domains and hence on the device functionality, since the transport through the amorphous regions is clearly the rate-limiting step.

Interestingly, our experimental results suggest a different scenario. From our analysis of the 2DES spectra, we gain rather precise information about the energy levels of the vibronically coupled exciton and polaron pair states, the strength of the vibronic coupling and the homogeneous and inhomogeneous broadening of the corresponding optical resonances. When using this information from the nonlinear measurements to simulate the linear optical spectrum of our P3HT thin films, we find that we can now describe it over the entire energy range between 1.9 and 3.0 eV (Fig. 3f). Importantly, no distinction between excitons in crystalline and amorphous regions is necessary in this simulation. Instead only one type of excitons (and polaron pairs) is assumed that is subject to a disordered potential with amplitude of ∼90 meV. This in fact implies that the excitons effectively average over the disordered potentials in both crystalline and amorphous regions and that the effect of the resulting averaged potential on the optical spectra may be modelled in terms of a single inhomogeneous line broadening parameter. Such a scenario is indeed known from other types of disordered, more delocalized Wannier-type excitonic systems[54].

When combined, our experimental and theoretical results strongly suggest that the elementary optical excitations of a prototypical conjugated polymer thin film at room temperature are strongly mixed exciton-polaron-pair-vibrational modes. Due to the interplay of electronic and vibronic couplings, charge-separated polaron pairs are formed within less than one vibrational period. Our model simulations (Supplementary Fig. 12) show that strong coherent vibronic couplings in this system significantly accelerate the initial photoinduced charge-separation dynamics, even in the presence of sizeable static disorder (Supplementary Fig. 13). We find for this system that the electronic coupling strength $J$ between exciton and polaron pair states, their energetic detuning, the energy of the coupled vibrational mode and the vibronic couplings are of the same order of magnitude, whereas the relaxation rate of the vibrational mode is much slower. In this case, an impulsive optical excitation of the exciton state results in persistent population oscillations between exciton and polaron pair states (see Supplementary Note 9). The strong vibronic coupling gives rise to state mixing and hence peak splittings in the 2DES maps. Our results suggest that such effects may also be present in many other systems with similarly strong electron-phonon couplings to underdamped vibrational modes. This supports predictions of recent ab-initio quantum dynamics simulations of light-induced charge transfer processes in donor-acceptor blends used in organic solar cell[24,33,55].

In summary, our results provide new insight into the initial quantum dynamics of excitons and polaron pairs in P3HT and present experimental evidence for coherent charge oscillations between both types of excitations. Although our theoretical analysis uses a model dimeric system, in the investigated samples such strong vibronic couplings are likely to extend over different segments of the disordered polymer chains. Surprisingly, we find that this vibronic model can describe the whole static absorption spectrum of our samples without any assumption on the film microstructure but only introducing inhomogeneous broadening. This suggests that our experiments do not just probe the optical properties of excitons in semi-crystalline regions of the sample but rather the average properties of excitons in both crystalline and amorphous regions and their common disorder potential. This could have interesting implications for the excitonic transport properties since it would mean that vibronic couplings induce coherent transport in both crystalline and amorphous regions and hence favour long-range transport. As such, these couplings may give rise to an initial ultrafast coherent transport of vibronic wavepackets along the polymer chains[32], with wavepacket coherence persisting for several hundreds of femtoseconds at room temperature. This finding may be of relevance for device applications because it opens up a new perspective for the optimization of charge transport in organic semiconductors by controlling vibronic coherence[56]. Since the

efficiency of organic-based devices like solar cells is often limited by the short diffusion lengths in conjugated polymers, achieving sizeable coherent transport lengths may significantly boost the performance. Our results suggest that tuning of electronic and vibronic couplings via chemical synthesis may allow for controlling the yield of photoinduced charge formation and hence, on a longer perspective, for optimizing the performance of organic-based optoelectronic devices.

## Methods

**Samples preparation.** Thin films of regioregular P3HT (Rieke Metals Inc.) were prepared by spin coating from a hot chlorobenzene solution on thin 180 μm glass substrates. The samples were subsequently annealed at 150 °C for 10 min. The entire sample preparation was carried out in a nitrogen filled glovebox. In order to prevent degradation, the polymer thin films were capped with a 150 nm LiF layer. No signs of photodegradation were observed during the measurements.

**Experimental set-up.** We recorded 2DES maps using an ultrafast home-built spectrometer based on a partially collinear configuration. A detailed scheme of the set-up and the chosen pulse sequence are depicted in Supplementary Fig. 1a,b, respectively. In our set-up, a home-built noncollinear optical parametric amplifier, pumped by a regeneratively amplified Ti:Sapphire laser operating at 1-kHz repetition rate, is used to generate ultrabroadband pulses with a spectrum ranging from 1.8 to 2.3 eV (Supplementary Fig. 1c). The pulses are compressed to <8 fs, estimated from the second harmonic frequency-resolved optical gating map (Supplementary Fig. 1d), with chirped mirrors and employed for both excitation and detection in a one-colour 2DES experiment. To this aim, the pulses are split into pump and probe pulses using a broadband beamsplitter. In the pump arm, a pair of phase-locked collinearly propagating pulses is generated in a highly stable interferometer based on birefringent α-BBO wedges, known as Translating-Wedge-Based Identical Pulses eNcoding System[57,58] (Supplementary Fig. 1a). The Translating-Wedge-Based Identical Pulses eNcoding System allows us to control the interpulse delay τ, known as the coherence time, with a precision of 10 as. The dispersion introduced by the wedges is compensated with an additional pair of chirped mirrors in the pump path. The probe pulses are time delayed by the waiting time T with a conventional translation stage. Pump and probe pulses are focused to a spot size of 50 μm onto the sample at a small angle using all-reflective optics. The probe pulses transmitted through the sample are spectrally dispersed in a monochromator and recorded with a 1,024-element photodiode array (Entwicklungsbüro Stresing). The transmitted pump pulses are blocked. Experimentally, we record normalized differential transmission spectra $\Delta T(\tau,T,E_D) = \frac{T_{on}(\tau,T,E_D) - T_{off}(\tau,T,E_D)}{T_{off}(\tau,T,E_D)}$ as differences of the probe transmission spectra $T_{on,off}(\tau,T,E_D)$ with the pump pulses being switched on or off, respectively. For each waiting time, $\Delta T$ spectra are recorded at ∼1,000 consecutive coherence times with a step size of 0.3 fs. The waiting time is scanned with a stepsize of 2.5 or 5 fs, respectively. The data are averaged over five consecutive scans of the waiting time. During each measurement, a calibration of the coherence time axis is performed by recording an interferogram of the pump pulses[58]. Absorptive 2D maps $A_{2D}(E_X,T,E_D)$ are calculated from the $\Delta T$ spectra at each waiting time by taking the real part of the Fourier transform along the coherence time axis τ,

$$A_{2D}(E_X,T,E_D) = \text{Re}\left\{\int_0^\infty \Delta T(\tau,T,E_D)e^{-i\frac{E_X}{\hbar}\tau}d\tau\right\} \qquad (1)$$

To avoid ringing in the 2D spectra, the interferograms $\Delta T(\tau)$ are multiplied with a Gaussian filter $G(\tau) = \exp\left(-4\ln(2)\tau^2/\tau_F^2\right)$ with $\tau_F = 120$ fs. For further noise suppression, the 2D spectra are smoothed by applying a moving average filter with a width of 5 meV along both the excitation and detection axes. This limits the energy resolution in the present experiments to ∼5 meV. The pulse sequence used in our experiments is sketched in Supplementary Fig. 1b. The pump pulse pair, delayed by the coherence time τ, interacts with the sample. After the waiting time T, the noncollinear probe pulse generates a third order sample polarization $P^{(3)}$. The field reemitted by this third order polarization then interferes with the transmitted probe laser field and is recorded in the differential transmission spectrum.

**Data availability.** The data that support the findings of this study are available from the authors upon reasonable request.

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

## Acknowledgements

Financial support by the European Union project CRONOS (Grant number 280879-2), PAPETS (Grant number 323901), QUCHIP (FETPROACT-3-2014: Quantum simulation, Grant number 641039), the Deutsche Forschungsgemeinschaft (SPP1391, SFB TRR/21, GRK 1885/1), the Korea Foundation for International Cooperation of Science and Technology (Global Research Laboratory project, K20815000003), the German-Israeli Foundation (GIF grant no. 1256), the MC-IIF MODENADYNA (grant agreement no. 623413) and the Italian FIRB ('Flashit' Project) is gratefully acknowledged. G.C. acknowledges financial support by the European Research Council (ERC-2011-AdG No. 291198). M.B.P. acknowledges financial support by the European Research Council (ERC-2012-Syn No. 319130) and an Alexander von Humboldt Professorship. This work was performed on the computational resource bwUniCluster funded by the Ministry of Science, Research and the Arts Baden-Württemberg and the Universities of the State of Baden-Württemberg, Germany, within the framework program bwHPC. C.L. wishes to thank I. Burghardt, D. Neher and F.C. Spano for helpful discussions.

## Author contributions

C.L., G.C. and E.M. initiated this work. A.D.S. prepared the samples. A.D.S., M.M., E.S. and J.R. performed the ultrafast spectroscopy experiments. A.D.S., E.S. and J.R. analysed the experimental data. A.D.S., F.T., C.L., C.A.R., J.L., S.F.H. and M.B.P. performed the theoretical analysis. C.L., A.D.S. and E.S. designed the paper. All authors discussed the results and contributed to the writing of the paper.

## Additional information

**Competing financial interests:** The authors declare no competing financial interests.

