## [Peer Review File · Nature Communications]

Reviewers' comments:

Reviewer #2 (Remarks to the Author):

I am fully satisfied by the answers to my comments. It is a tough call with the referee 1 – I really do not have enough time to go through all the details of the discussion (it is really long). I did read it and found the answers convincing and in my opinion warranting publication in Nature Comm.

Reviewer #3 (Remarks to the Author):

By and large, the manuscript is enough high-interest to be recommended for publication. However, the authors should take care of the following points, introduced in order of importance:

1. The authors have, in my opinion, still not satisfactorily addressed the issue of order/disorder in their P3HT samples. They still seem to imply that the chains are generally disordered but annealed P3HT is expected to have a high degree of (semi)-crystallinity. How can the authors be sure that what they see predominantly does not come from the crystalline regions?

This aspect is also relevant in the context of their statement in the Conclusions that: "This finding may be of relevance for device applications because it opens up a new perspective for the optimization of charge transport in organic semiconductors by controlling vibronic coherence. Since the efficiency of organic-based devices like solar cells is often limited by the short diffusion lengths in conjugated polymers, achieving sizable coherent transport lengths may significantly boost the performance." If this were proven to happen truly in a disordered system, it would indeed be significant...

2. There should be a more explicit discussion of the (static) model of Spano (for instance, in the case of oligocene crystals), in the sense that what they report is a coherent superposition of a Frenkel [i.e. local] exciton and a charge-transfer [i.e. polaron pair] exciton.

3. I am not fond of the first 3 references of the paper. Proper credit should rather be given to those researchers who initially developed those applications (for instance, for OLEDs, Ching Tang and the Friend group).

REVIEWERS' COMMENTS:

Reviewer #3 (Remarks to the Author):

The authors have adequately addressed my remaining concerns. My recommendation is that the paper be accepted.

Reviewers' comments:

Reviewer #2 (Remarks to the Author):

I am fully satisfied by the answers to my comments. It is a tough call with the referee 1 – I really do not have enough time to go through all the details of the discussion (it is really long). I did read it and found the answers convincing and in my opinion warranting publication in Nature Comm.

Answer: we are very pleased to read that our response “fully satisfies” the Reviewer and that he/she recommends publication of our revised manuscript in Nature Communications. We wish to thank him/her for the time and the valuable comments and suggestions.

Reviewer #3 (Remarks to the Author):

By and large, the manuscript is enough high-interest to be recommended for publication. However, the authors should take care of the following points, introduced in order of importance: 1. The authors have, in my opinion, still not satisfactorily addressed the issue of order/disorder in their P3HT samples. They still seem to imply that the chains are generally disordered but annealed P3HT is expected to have a high degree of (semi)-crystallinity. How can the authors be sure that what they see predominantly does not come from the crystalline regions? This aspect is also relevant in the context of their statement in the Conclusions that: "This finding may be of relevance for device applications because it opens up a new perspective for the optimization of charge transport in organic semiconductors by controlling vibronic coherence. Since the efficiency of organic-based devices like solar cells is often limited by the short diffusion lengths in conjugated polymers, achieving sizable coherent transport lengths may significantly boost the performance." If this were proven to happen truly in a disordered system, it would indeed be significant...

Answer: We are glad to read that the Reviewer finds our revised manuscript of enough “high-interest” to recommend it for publication. We are also thankful for the insightful questions which we will address in detail below.

We agree with the Reviewer that “P3HT is expected to have a high degree of (semi)-crystallinity”, meaning by that a microstructure exhibiting at the same time highly ordered domains and completely amorphous regions. This picture is confirmed by several studies of this model polymer that have addressed the critical task of understanding and possibly controlling the morphology-function relation of thin films [H. Sirringhaus et al, Nature 1999; A. Salleo et al, Advanced materials 2010; G. Li et al, Advanced functional materials 2007; A. Zen et al, Advanced functional materials 2004; S. Ludwigs ed, P3HT revisited, Springer 2014; ...]. From these studies it is known that not only characteristic parameters like the average molecular weight, but also the processing conditions (e.g. solvent and annealing) have an impact on the morphology. In particular, all these parameters may change the amount and size of crystallites and amorphous domains [S. Ludwigs ed, P3HT revisited, Springer 2014]. It also seems more or less accepted that this interplay between crystalline and amorphous regions affects the linear optical spectra of P3HT in the sense that at least two types of excitons, i.e. one at lower energies from the crystallites and one at higher energies from the

amorphous regions, contribute to these spectra. The absorption spectrum of excitons in crystalline regions is formally described by the “Spano model” [Clark et al PRL 2007], which nicely reproduces the low energy part of the absorption spectrum of P3HT thin films, but not the high energy part. According to the literature [J. Clark et al, APL 94, 163306, 2009; S. Ludwigs ed, P3HT revisited, Springer 2014], the discrepancy is due to the absorption of excitons from amorphous regions which should result in a higher energy, unstructured band in the linear absorption, that is not accounted for in the model.

Based on this model one may indeed expect that the persistent vibronic coupling phenomena that are seen in our experiments can mainly be assigned to excitons in crystalline domains whereas the amorphous regions may act as blocking layers suppressing transport between adjacent crystalline domains. In such a scenario, any type of vibronic quantum coherence at early times may have little effect on the transport across crystalline domains and hence on the device functionality, since the transport through the amorphous regions is clearly the rate-limiting step.

Interestingly, our experimental results suggest a different scenario. From our analysis of the 2DES spectra, we gain rather precise information about the energy levels of the vibronically coupled exciton and polaron pair states, the strength of the vibronic coupling and inhomogeneous broadening of the corresponding optical resonances. When using this information from the nonlinear measurements to simulate the linear optical spectrum of our samples (cf. Fig. 3f) we find that we can now reproduce the linear absorption spectrum of our P3HT thin films over the entire energy range between 1.9 and 3.0 eV, i.e. both in the low and high energy part of the absorption spectrum. Importantly, no distinction between excitons in crystalline and amorphous regions is made in this simulation. Instead only one type of excitons (and polaron pairs) is assumed that is subject to a disordered potential with an amplitude of ~ 90 meV. This in fact implies that the excitons effectively average over the disordered potentials in both crystalline and amorphous regions and that the effect of the resulting averaged potential on the optical spectra may be modeled in terms of a single inhomogeneous line broadening parameter. Such a scenario is indeed known from other types of disordered, more delocalized Wannier-type excitonic systems [Intonti et al., PRL 2001]. This suggests that our experiments do not just probe the optical properties of excitons in semi-crystalline regions of the sample but rather the average properties of excitons in both crystalline and amorphous regions and that these excitons are subject to a common disorder potential. This could have interesting implications for the excitonic transport properties since it would mean that electron-phonon coupling could induce coherent vibronic transport over both crystalline and amorphous regions and hence favor long-range transport.

The experiments presented in this work do not directly relate the optical signatures to either crystalline or amorphous regions. Hence additional measurements probing the correlation between optical spectra and device morphology are certainly needed to validate the conjectures discussed above. With scattering type near-field optical spectroscopy now routinely approaching or beating the 10-nm resolution barrier, local nano-optical spectroscopy may be very helpful in gaining this information. Also, nano-optical photocurrent experiments may provide important new insight into the interplay between optical and transport properties. Such experiments are currently underway in our laboratories.

Changes: we have now added the following text to the Discussion section. Changes in the manuscript are highlighted in red.

„ P3HT is expected to have a high degree of semi-crystallinity, that is a microstructure exhibiting at the same time highly ordered domains and completely amorphous regions. This picture is confirmed by several studies of this model polymer that have addressed the critical task of understanding and possibly controlling the morphology-function relation of thin films^{38, 49, 50, 51, 52}. From these studies it is known that not only characteristic parameters like the average molecular weight, but also the processing conditions (e.g. solvent and annealing) have an impact on the morphology. In particular, all these parameters may change the amount and size of crystallites and amorphous domains³⁸. It is also a frequent claim in the literature that this interplay between crystalline and amorphous regions affects the linear optical spectra of P3HT in the sense that at least two types of excitons, i.e one at lower energies from the crystallites and the other one at higher energies from the amorphous regions, contribute to the spectra. The absorption spectrum of excitons in crystalline regions is formally described by the exciton model introduced by Spano³⁹, which nicely reproduces the low energy part of the absorption spectrum, but not the high energy part (cf. Fig. 1a). According to the literature^{38, 53}, the discrepancy is due to the absorption of excitons from amorphous regions which should result in a higher energy, unstructured band in the linear absorption that is not accounted for in the model. Based on this model³⁹ one may expect that the persistent vibronic coupling phenomena observed in our experiments can mainly be assigned to excitons in crystalline domains, whereas the amorphous regions may act as blocking layers suppressing charge and excitation transport between adjacent crystallites. In such a scenario any type of vibronic quantum coherence at early times may have little effect on the transport across crystalline domains and hence on the device functionality, since the transport through the amorphous regions is clearly the rate-limiting step.

Interestingly, our experimental results suggest a different scenario. From our analysis of the 2DES spectra, we gain rather precise information about the energy levels of the vibronically coupled exciton and polaron pair states, the strength of the vibronic coupling and the homogeneous and inhomogeneous broadening of the corresponding optical resonances. When using this information from the nonlinear measurements to simulate the linear optical spectrum of our P3HT thin films, we find that we can now describe it over the entire energy range between 1.9 and 3.0 eV (Fig. 3f). Importantly, no distinction between excitons in crystalline and amorphous regions is necessary in this simulation. Instead only one type of excitons (and polaron pairs) is assumed that is subject to a disordered potential with amplitude of ~ 90 meV. This in fact implies that the excitons effectively average over the disordered potentials in both crystalline and amorphous regions and that the effect of the resulting averaged potential on the optical spectra may be modeled in terms of a single inhomogeneous line broadening parameter. Such a scenario is indeed known from other types of disordered, more delocalized Wannier-type excitonic systems⁵⁴.

In summary, our results provide fundamentally new insight into the initial quantum dynamics of excitons and polaron pairs and present experimental evidence for coherent charge oscillations between both types of excitations. Although our theoretical analysis uses a model dimeric system, in the investigated samples such strong vibronic couplings are likely to extend over different segments of the disordered polymer chains. Surprisingly, we find that this vibronic model can describe the whole static absorption spectrum of our samples without any assumption on the film microstructure but only introducing inhomogeneous broadening. This suggests that our experiments do not just probe the optical properties of excitons in semi-crystalline regions of the sample but rather the average properties of excitons in both crystalline and amorphous regions and their common disorder potential. This could have interesting implications for the excitonic transport properties since it

would mean that vibronic couplings induce coherent transport in both crystalline and amorphous regions and hence favor long-range transport.”

2. There should be a more explicit discussion of the (static) model of Spano (for instance, in the case of oligocene crystals), in the sense that what they report is a coherent superposition of a Frenkel [i.e. local] exciton and a charge-transfer [i.e. polaron pair] exciton.

Answer: Following the suggestion of the Reviewer, we have now added a short paragraph in the manuscript where we explicitly mention the model of Spano for the description of oligoacene crystals. Nevertheless, we would like to underline that this is a static model used to explain linear optical absorption spectra, very much like the “Spano” model that has previously been used to explain the linear optical spectra of P3HT [Clark et al., PRL 2007]. Microscopic models for dephasing and relaxation have not been explicitly included and hence the model cannot directly predict exciton and polaron pair dynamics in the system. Also, an excitonic “Spano” model, neglecting vibronic coupling between excitons and polaron pairs is commonly used to explain the linear optical spectra of P3HT [Clark et al., PRL 2007].

Change: “A similar model allowing for a coherent superposition of localized Frenkel and charge transfer excitons has been proposed to model the static absorption spectrum of oligoacene crystals⁴⁷, whereas, commonly, this superposition is not considered in models of the static absorption of P3HT³⁹.”

3. I am not fond of the first 3 references of the paper. Proper credit should rather be given to those researchers who initially developed those applications (for instance, for OLEDs, Ching Tang and the Friend group).

Answer: We thank the Reviewer for the advice. Our initial attention was not to give reference to the initial developments but rather to highlight important recent developments in the fields. We have now added appropriate references to initial developments in all three applications that are listed in the paper.

Change: The first sentence of the introduction now reads: “Thin films of conductive polymers combine mechanical flexibility with excellent optical and electronic properties and hence are key active materials for flexible organic optoelectronic devices such as solar cells [Yu 1995, Li 2012], field effect transistors [Koezuka 1987, Horowitz 1998, Sekitani 2010] or light emitting diodes [Tang 1987, Burroughes 1990, Reineke 2009]”.